# Isolation of Clinical Microbial Isolates during Orthodontic Aligner Therapy and Their Ability to Form Biofilm

**DOI:** 10.3390/dj11010013

**Published:** 2023-01-03

**Authors:** Oleg Baybekov, Yaroslav Stanishevskiy, Nadezhda Sachivkina, Anna Bobunova, Natallia Zhabo, Marina Avdonina

**Affiliations:** 1Institute of Biochemical Technology and Nanotechnology, Peoples Friendship University of Russia (RUDN University), 117198 Moscow, Russia; 2Department of Microbiology V.S. Kiktenko, Peoples Friendship University of Russia (RUDN University), 117198 Moscow, Russia; 3Department of Foreign Languages, Institute of Medicine, Peoples’ Friendship University of Russia (RUDN University), 117198 Moscow, Russia; 4Department of Linguistics and Intercultural Communication, Moscow State Linguistic University, 119034 Moscow, Russia

**Keywords:** bacteria, biofilms, dental abutments, dental aligners, microbiology, optical density, microscopy, phenotypic features

## Abstract

The purpose of this study is to calculate microbiological composition of aligners after a day of wearing them. To date, the dental market for orthodontists offers many ways to correct bites. Aligners are transparent and almost invisible from the teeth. They are used for everyday wear to correct the incorrect position of the teeth, which was once considered the prerogative of braces. Scientists worldwide have repeatedly considered questions regarding the interaction between aligners and the oral cavity’s microflora; however, the emphasis has mainly shifted toward species composition and antibiotic resistance. The various properties of these microorganisms, including biofilm formation, adhesion to various cells, and the ability to phagocytize, have not been studied so widely. In addition, these characteristics, as well as the microorganisms themselves, have properties that change over time, location, and in certain conditions. In this regard, the problem of biofilm formation in dental practice is always relevant. It requires constant monitoring since high contamination of orthodontic materials can reduce the effectiveness of local anti-inflammatory therapy and cause relapses in caries and inflammatory diseases of the oral cavity. Adhesive properties, one of the key factors in forming the architectonics of biofilms, provide the virulence factors of microorganisms and are characterized by an increase in optical density, determining the duration and retrospectivity of diagnostic studies. This paper focuses on the isolation of clinical microbial isolates during aligner therapy and their ability to form biofilms. In the future, we plan to use the obtained strains of microorganisms to create an effective and safe biofilm-destroying agent. We aimed to study morphometric and densitometric indicators of biofilms of microorganisms persisting on aligners.

## 1. Introduction

The oral cavity constantly encounters microorganisms [1]. Plaque biofilm, the main cause of caries, periodontitis, and other dental diseases, is a complex community of bacteria and fungi that causes infection by protecting pathogenic microorganisms from external medicinal agents and avoiding host defense mechanisms [2]. Although a significant amount of research is aimed at developing antimicrobial agents to solve this problem, most have low efficacy and safety [3,4].

R.G. Gibbons et al. were the first to investigate the interaction between representatives of oral microflora and their adhesion to the surface of filling materials [5,6]. An important factor in the adhesion of microorganisms to the surface of the filling material and enamel is the presence of saliva, which stimulates this adhesion and contributes to the development of caries [7]. Moreover, the concentration of saliva glycoproteins positively correlates with the adhesion of *S. mutans* on the surface of enamel and dental fillings; that is, the higher the concentration of saliva glycoproteins, the higher the probability of *S. mutans* adhesion. As a result of research [8,9], R.G. Gibbons suggested that *S. sanguis* adheres to composite fillings better than *S. mutans*. However, this does not correspond to the main hypothesis that *S. mutans* is the most virulent (cariogenic) representative with the best adhesive ability due to its production of insoluble glycans, which improves these microorganisms’ attachment to the seal’s surface [10,11].

Since the appearance of transparent aligners, a device for light and moderate orthodontic movement of teeth, in 1998, the problem of biofilm formation has become relevant. Transparent aligner therapy has been a part of orthodontic practice for decades and is becoming an increasingly common addition to the orthodontic arsenal. Internet searches show hundreds of articles currently devoted to the problem of biofilm formation during orthodontic treatment. This study is the first of a whole series of studies, we decided not to focus on the difference in the composition of biofilms in the first article. In the next article there will be a different age of the participants and we will compare the results, focus on strains and differences in biofilms.

Fixed and removable orthodontic devices can affect the composition of oral microbiota through two mechanisms: (1) accumulation of plaque and (2) violation of oral hygiene. Quantitative and qualitative changes in the oral cavity’s microbiota associated with orthodontic treatments, since they cover an extensive surface of the teeth, are reflected in many scientific papers [12,13,14,15]. Accumulation of food residues, bacterial plaque, and increasing difficulty in maintaining good oral hygiene by patients are the main risk factors for the occurrence or exacerbation of dental caries and gingivitis/periodontitis [16,17,18].

We aimed to study morphometric and densitometric indicators of biofilms persisting on aligners.

## 2. Materials and Methods

### 2.1. Microbial Strains

During the study, a microbiological examination of 10 persons using aligners obtained from adolescents aged 13 to 19 years was conducted (six female and four male participants). The choice of this age group is due to the high prevalence of this orthodontic procedure compared to people of other groups. In further work, we plan to repeat the studies in other age groups, including 20–29 and 30–39 years. At a more mature age, wearing orthodontic aligners is not so popular. The procedure is not recommended for persons under the age of 13, and is only for special purposes.

Patients used Nuvola^®^ aligners. They are made of polyethylene terephthalate glycol (PETG), a light, resistant, and very clear material. We asked patients not to wash the aligners after a day of wearing them. In the clinic, patients received a new pair of aligners, and the old one was used in the experiment. These aligners with biofilms of microorganisms were sliced and placed in sterile wide test tubes filled with 10 mL of sterile 0.85% NaCl solution. For the complete removal of microorganisms, the test tubes were left in the solution for 3 h and then thoroughly mixed in the vortex for 10 min (3000 rpm) (Figure 1). A total of 1 mL of the sample was obtained from a test tube.

Then, we prepared serial dilutions, plated the diluted suspensions, and counted the number of colony-forming units. For yeast-like fungi (YLF), Sabouraud dextrose agar (BioMerieux, France) was used. For *Bifidobacterium* spp., Blaurock medium (HEM, Moscow, Russia) was used. For *Lactobacillus* spp., MRS medium (HiMedia, India) was used. For *Staphylococcus* spp., peptone–salt medium and yolk–salt agar (HiMedia, India) were used. For *Streptococcus* spp., mitis-salivarius agar (HiMedia, India) was used. For the *Enterobacteria family*, Endo’s medium, Ploskirev’s medium, and bismuth-sulfite agar (HEM, Moscow, Russia) were used. For *Clostridium* spp., clostridial agar (HiMedia, India) was used, and for all others, meat-peptone agar (MPA) (HEM, Moscow, Russia) was used. The plates were incubated at 37–38 °C for 24–72 h. We used AnaeroJar anaerostat and Anaerocult gas-generating packages (Merk, Germany) for 72 h at 37 °C to create anaerobic conditions. The pure cultures were identified using a matrix-activated laser desorption/ionization technology by MALDI Biotyper (Bruker Daltonik Inc., Billerica, MA, USA). The values of the X score ranged from 0 to 3, and values from 2 to 3 were considered successful. A score of >2.3 was considered highly reliable [19,20].

The number of microorganisms (contamination index—*C*) in 1.0 cm^3^ of the sample was calculated using the formula and expressed as a logarithm with base 10 (lg CFU/1 mL, where CFU is colony forming unit):C=(N/V)×K
where *N* represents the average number of colonies in one bacteriological cup, *V* represents the volume of suspension applied when seeding the surface of the agar, and *K* represents the multiplicity of dilution [21,22].

If different scores were detected by MALDI Biotyper, the culture of microorganisms was considered a separate strain. We used established guidelines CLSI for microbiological identification by MALDI-TOF MS (https://clsi.org/standards/products/microbiology/documents/m58/, accessed on 6 December 2022) and database MALDI Biotyper (Bruker Daltonics GmbH, Bremen, Germany) [23,24]. The number of different strains from the same species was calculated in absolute numbers (A.n.). The percentage of strains was also entered in the table.

Microorganism cultures were stored in semi-liquid 0.5% of meat-peptone agar in freeze-dried form at 4 ± 1 °C.

### 2.2. Densitometric Indicators of Microbial Biofilms

Microbial biofilms were indicated by the degree of crystal violet binding (HiMedia, India) at 490 nm wavelength. The tested samples were added to the wells of a 96-well plate (Medpolymer Company, Russia), and cultivated in a constant aerobic environment of 37 °C for 48 h. The liquid was discarded, and the wells were washed three times (pH 7.3) with 200 μL of phosphate-buffered solution (PBS). The plates were shaken for 5 min at each washing stage. The samples were fixed with 150 μL of 96% ethanol for 15 min and dried out at 37 °C for 20 min. The microbial biofilms were stained by adding 0.5% stain solution to each well and subsequent cultivation at 37 °C for 5 min. The contents of the wells were discarded, and the plates were washed three times (pH 7.3) with 200 μL of PBS and dried out. The bound stain was eluted from the attached cells with 200 μL of 96% ethanol for 30 min [25,26].

The optical density (OD) of biofilms was determined by the degree of crystal violet binding [27].

### 2.3. Confocal Laser Scanning Microscopy

The absence of biofilms on the aligners after washing was analyzed by the CSLM LSM510/ConfoCor2 system (Carl Zeiss, Oberkochen, Germany). Pieces of aligners were stained with concanavalin A (50 mg/L) (Molecular Probes, Eugene, OR, USA) at 37 °C for 1 h (green fluorescence). Approximately 100 sections were created from the studied surfaces. The absence of a fluorescent signal proved the absence of biofilms on the aligners.

### 2.4. Statistics

We analyzed the results using SPSS 20.0 (IBM Corp., Armonk, NY, USA). The significance of the results was determined using Student’s *t*-test and when *p* < 0.05.

## 3. Results and Discussions

Microorganisms belonging to 28 species were isolated and identified by studying the oral microflora in the observed group (Table 1). Studying the contamination index helped compare groups of microorganisms according to the degree of dominance and identify those most important to the microecosystem formation of aligners. The data analysis in Table 1 shows that the most important representatives (≥5 lg CFU/1 mL) of oral microflora when teenagers wear aligners are *bifidobacteria*, YLF of the genus *Candida—Candida albicans*, *E. coli*, *Peptostreptococcus anaerobius*, *Porphyromonas gingivalis*, *Prevotella buccae*, *Staphylococcus aureus*, and *Streptococcus mitis*. We detected 13 Gram-positive and 13 Gram-negative bacteria, indicating the equivalent participation of these bacteria in the cell wall structure while forming plaque microbiota. The digital expression of ecological significance in other microflora representatives is significantly less than the values established for the dominant species (<5 lg CFU/1 mL), indicating their insignificant contribution to the structure of biofilm biocenoses.

*C. albicans*’ colonization of the mucous membranes may result in the acquisition and preservation of a stable population of Candida, preventing clinical infection development. The higher YLF concentration can be explained by the pathogenicity factors of *C. albicans*, which can be conditionally divided into five groups. However, when pathology occurs in the body, their effects are conducted simultaneously:

1. The ability to adhere to host tissues is the first step to interaction with a microorganism;

2. Production of proteolytic enzymes—secretory aspartyl proteases (SAP), which facilitate the penetration and invasion of Candida into tissues;

3. Morphological transformation of the “yeast–hyphal form”, which also facilitates the penetration of YLF into tissues and helps the microorganism bypass the host’s defense systems;

4. Various immunomodulatory actions (mechanisms) of some *C. albicans* molecules that may contribute to reducing the effectiveness of antifungal immunity;

5. Phenotypic switching characteristic of Candida when the conditions of existence change.

## Morphometric and Densitometric Indicators of Biofilms

Biofilm formation represents the species’ ability to occupy a dominant position in a community and exert a predominant influence on biocenotic processes, which determine the type of biocenosis according to the dominant ecological groupings [28,29,30,31]. In this regard, it was interesting to evaluate the community structure of bacteria and fungi and determine the degree of biofilm formation for each isolate within the species (Table 2). The strongest producers of biofilms were (OD_S_ ≥ 0.3): *Actinomyces israelii*, *Actinomyces naeslundii* isolate 3, *Campylobacter concisus*, *Candida albicans*, *Candida parapsilosis*, *Capnocytophaga gingivalis* isolate 1, *Escherichia coli*, *Fusobacterium nucleatum*, *Lactobacillus rhamnosus*, *Peptostreptococcus anaerobius*, *Prevotella buccae*, *Prevotella denticola*, and *Staphylococcus aureus.* All other community types on the liners had biofilm formation rates many times lower. All isolates with a high ability to form biofilms were stored for future use in a freeze-dried form at 4 ± 1 °C.

Although orthodontic treatment with aligners has shown encouraging results in terms of plaque index and gum condition control compared to classical fixed orthodontic treatment [32,33,34], bacteria can form oral biofilms on the surface of clear aligners. Tektas et al. [35] demonstrated that the initial microbial adhesion and biofilm formation of aerobic and anaerobic oral cavity types were similar between enamel, metal orthodontic braces, and clear aligners. In addition, bacterial adhesion to aligners increases due to the shape of the aligner, which is not straight and contains grooves and protrusions. In addition, Low et al. [36] showed that on the surface of the aligner itself, even if it is new, there are micro-scratches, microcracks, and slight elevations. Bacterial and fungal biofilms attach to these irregularities. In addition, Gracco et al. [37] demonstrated physical changes on the aligners when wearing them. After 14 days, microcracks, and worn and delaminated areas appeared on them, promoting adhesion and bacterial growth, as well as localized deposits of calcined biofilm and loss of transparency. Schuster et al. [38] and our study analyzed used aligners and found the erasure of protrusions and adsorption of buccal epithelial cells mixed with microbial biofilms. Finally, Low et al. [36] studied the ultrastructure and morphology of biofilms on aligners. They found that the initial biofilm consisted of mostly coccal bacterial species, including streptococci and staphylococci. This biofilm included Gram-negative flora and, after a while, fungi. A similar experiment was conducted by Zhao et al. [39] who studied microbial changes in the oral cavity during aligner treatment, comparing the contamination indicators in 25 patients before and after six months. They revealed the appearance of periodontal pathogens and cariogenic bacteria, including *Aggregatibacter actinomycetemcomitans*, *Fusobacterium nucleatum*, *Treponema denticola*, *Porphyromonas gingivalis*, *Streptococcus mutans*, and *Streptococcus sobrinus*. They concluded that more long-term, high-quality investigations are necessary to elucidate whether changes in oral and periodontal microbiology associated with the placement of orthodontic appliances return completely to pretreatment levels.

In interesting research of Sfondrini M.F. et al. [40], periodontal status and microbiological composition of oral microbiota induced by clear aligner treatment were measured. A total of 20 orthodontic patients were submitted to professional oral hygiene and, subsequently, underwent aligners. They proved that aligner therapy does not significantly affect periodontal and microbiological parameters with respect to untreated patients for the first two months of therapy.

## 4. Conclusions

Our results proved that morphometric and densitometric indicators of heterogeneous biofilms could be used in developing antibacterial drugs for patients undergoing orthodontic treatments. We found 13 Gram-positive and 13 Gram-negative bacteria, and 2 yeast-like fungi. We tested isolated microorganisms’ ability to produce biofilms and identified the strongest producers. There are currently no means of ensuring the direct and complete destruction of biofilms. However, when we revealed the pathogenetic mechanisms behind the microorganisms’ relationship in biofilms, we developed an understanding of how to create and develop new drugs. Our study had several limitations, including the small sample of 10 patients and only participants of the 13–19 age category. Future studies should consider other age categories and groups of more than 10 people. Additionally, we plan to use the obtained strains of microorganisms to create an effective and safe biofilm-destroying agent.

## Figures and Tables

**Figure 1 dentistry-11-00013-f001:**
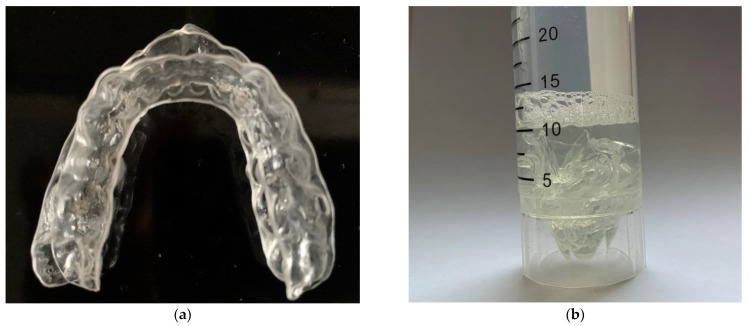
(**a**) used aligners from one patient; (**b**) aligners sliced and placed in a solution.

**Table 1 dentistry-11-00013-t001:** Groups of microorganisms according to the degree of dominance on aligners.

Species	Gram-Negative (−) or Positive (+)	Number of Isolates		
		lg CFU/1 mL	Absolute Number	%
*Actinomyces israelii*	+	2.45 ± 1.01	2	2.94
*Actinomyces naeslundii*	+	4.54 ± 1.38	3	4.41
*Actinomyces viscosus*	+	2.29 ± 0.75	1	1.47
*Bifidobacterium bifidum*	+	5.67 ± 2.21	4	5.88
*Campylobacter concisus*	−	2.35 ± 0.51	1	1.47
*Campylobacter gracilis*	−	3.22 ± 0.78	2	2.94
*Candida albicans*	YLF	6.38 ± 1.86	4	5.88
*Candida parapsilosis*	YLF	2.45 ± 0.60	3	4.41
*Capnocytophaga gingivalis*	−	4.05 ± 1.20	3	4.41
*Clostridium aminobutyricum*	+	1.19 ± 0.29	2	2.94
*Escherichia coli*	−	5.67 ± 2.05	5	7.35
*Fusobacterium canifelinum*	−	1.18 ± 0.43	2	2.94
*Fusobacterium nucleatum*	−	2.54 ± 0.86	2	2.94
*Gemella sanguinis*	+	2.66 ± 0.97	3	4.41
*Lactobacillus rhamnosus*	+	4.38 ± 1.27	2	2.94
*Leptotrichia buccalis*	−	4.82 ± 1.31	2	2.94
*Leptotrichia shahii*	−	2.16 ± 0.50	1	1.47
*Peptostreptococcus anaerobius*	+	8.72 ± 2.24	3	4.41
*Porphyromonas gingivalis*	−	6.40 ± 2.01	3	4.41
*Prevotella buccae*	−	5.68 ± 1.80	4	5.88
*Prevotella denticola*	−	2.89 ± 0.56	2	2.94
*Prevotella intermedia*	−	2.42 ± 0.84	1	1.47
*Prevotella oris*	−	1.39 ± 0.26	1	1.47
*Staphylococcus aureus*	+	6.05 ± 1.44	4	5.88
*Streptococcus gordonii*	+	3.88 ± 1.20	3	4.41
*Streptococcus mitis*	+	7.51 ± 1.64	3	4.41
*Streptococcus sanguinis*	+	0.27 ± 0.18	1	1.47
*Streptococcus salivarius*	*+*	0.31 ± 0.20	1	1.47
Total: 28	13 Gr+ and 13 Gr−		68	100%

CFU—colony forming unit; YLF—yeast-like fungi.

**Table 2 dentistry-11-00013-t002:** Determination of bacterial biofilm formation intensity by optic density.

	Optic Density						
	OD_S_					OD_A_	Average Error
	1	2	3	4	5		
*Actinomyces israelii*	0.312	0.331	np	np	np	0.3215	0.0095
*Actinomyces naeslundii*	0.244	0.293	0.302	np	np	0.2797	0.0238
*Actinomyces viscosus*	0.284	np	np	np	np	np	np
*Bifidobacterium bifidum*	0.246	0.256	0.267	0.275	np	0.261	0.01
*Campylobacter concisus*	0.358	np	np	np	np	np	np
*Campylobacter gracilis*	0.261	0.265	np	np	np	0.263	0.002
*Candida albicans*	0.420	0.502	0.431	0.384	np	0.4343	0.0338
*Candida parapsilosis*	0.390	0.347	0.333	np	np	0.3567	0.0222
*Capnocytophaga gingivalis*	0.350	0.291	np	np	np	0.3205	0.0295
*Clostridium aminobutyricum*	0.279	0.193	np	np	np	0.236	0.043
*Escherichia coli*	0.462	0.411	0.380	0.376	0.382	0.4022	0.0274
*Fusobacterium canifelinum*	0.177	0.193	np	np	np	0.185	0.008
*Fusobacterium nucleatum*	0.398	0.347	np	np	np	0.3725	0.0255
*Gemella sanguinis*	0.199	0.193	0.180	np	np	0.1907	0.0071
*Lactobacillus rhamnosus*	0.325	0.303	np	np	np	0.314	0.011
*Leptotrichia buccalis*	0.179	0.191	np	np	np	0.185	0.006
*Leptotrichia shahii*	0.208	np	np	np	np	np	np
*Peptostreptococcus anaerobius*	0.418	0.403	0.384	np	np	0.4017	0.0118
*Porphyromonas gingivalis*	0.132	0.176	0.173	np	np	0.1603	0.0189
*Prevotella buccae*	0.327	0.345	0.396	0.401	np	0.3673	0.0313
*Prevotella denticola*	0.470	0.392	np	np	np	0.431	0.039
*Prevotella intermedia*	0.132	np	np	np	np	np	np
*Prevotella oris*	0.193	np	np	np	np	np	np
*Staphylococcus aureus*	0.421	0.394	0.385	0.400	np	0.4	0.0105
*Streptococcus gordonii*	0.189	0.195	0.215	np	np	0.1997	0.0102
*Streptococcus mitis*	0.244	0.230	0.261	np	np	0.245	0.0107
*Streptococcus sanguinis*	0.281	np	np	np	np	np	np
*Streptococcus salivarius*	0.253	np	np	np	np	np	np

OD_S_—tested sample; OD_A_—average value; np—not possible.

## Data Availability

Data are contained within the article.

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
