# Peer review of "Isolation of Clinical Microbial Isolates during Orthodontic Aligner Therapy and Their Ability to Form Biofilm"

_dentistry, 2023, doi:10.3390/dj11010013_

Round 1
Reviewer 1 Report
Dear Authors,
Thank you for your work submitted to the journal.
I think that it is relevant from a clinical point of view.
However, some improvements are required before considering its publication.
In particular:
- the statistical null hypothesis should be clearly stated
- in the discussion, you should state whether the statistical null hypothesis has been accepted or not
- the discussion section should be implemented taking into account further recent literature. In particular, you could consider the following research: Sfondrini et al. Microbiological Changes during Orthodontic Aligner Therapy: A Prospective Clinical Trial. Appl. Sci. 2021, 11, 6758.
Author Response
Dear Reviewer! Thank you so much for paying attention to our work and spending your time. Our team very much appreciates your edits in the article and of course we will take them into account. We are sure that working together will only make the article better. We have tried to answer all your questions:
In particular:
- the statistical null hypothesis should be clearly stated
The null hypothesis is the theory that there are some two aggregates that do not differ from each other. However, at the scientific level there is no concept of "do not differ", but there is "their similarity is zero". From this definition, the concept was formed. In statistics, the null hypothesis is denoted as H0. Moreover, the extreme value of the impossible (unlikely) is considered to be from 0.01 to 0.05 or less.
- in the discussion, you should state whether the statistical null hypothesis has been accepted or not
The null hypothesis will state that there is no difference between the values of concentrations of microorganisms. An alternative hypothesis would say that there are significant differences in concentration. Please, if we have incorrectly formulated the answer to your question, please specify
- the discussion section should be implemented taking into account further recent literature. In particular, you could consider the following research: Sfondrini et al. Microbiological Changes during Orthodontic Aligner Therapy: A Prospective Clinical Trial. Appl. Sci. 2021, 11, 6758.
Thank you so much for the hint! We will definitely use this resource!!!
Reviewer 2 Report
In this article, authors have isolated microorganism responsible for biofilms in patients using aligners. Need some major clarifications before publication.
· Revise the abstract. Key finding should be highlighted in abstract. Irrelevant description should be removed.
· In section 2.1, did authors only collected one sample per patient? Secondly, why samples were only collected at 24, which is minimum time for biofilm growth?
· Ethical approval for human studies is missing.
· What is chemical composition of aligners? Do its composition have influence on type of biofilm growth?
· It would be interesting if authors bifurcate data in table on gender basis.
· Please comments on dominance of YLF in sampled biofilms. Do sampling time, type of aligners material, diet or age group have any role to play in it?
· Confocal laser scanning microscopy images and interpretation is missing.
· Conclusion need revision. Explain how your data is helpful in new drug development?
Author Response
Dear Reviewer! Thank you so much for paying attention to our work and spending your time. Our team very much appreciates your edits in the article and of course we will take them into account. We are sure that working together will only make the article better. We have tried to answer all your questions:
Revise the abstract. Key finding should be highlighted in abstract. Irrelevant description should be removed.
Of course, we have formulated the main goal of our research. On the other hand, abstract should not be repeated with conclusions. Unfortunately, another reviewer asked us not to shorten, but to expand the abstract. There is a conflict of interest. We respect and consider the opinion of each reviewer. Thanks
- In section 2.1, did authors only collected one sample per patient? Secondly, why samples were only collected at 24, which is minimum time for biofilm growth?
In section 2.1, we wrote that we took the used aligners from the patient: from the lower and upper jaw. Crushed them. Microbiological analysis was performed according to the international protocol. The incubation of microorganisms lasted from 24 to 72 hours. We didn't mention 24 hours anywhere. The biofilms were incubated for 48 hours.
- Ethical approval for human studies is missing.
In fact, there were no studies on humans or animals. We took used aligners from patients, which they intended to throw away. We have taken the consent from the patients that we will conduct microbiological studies with this material. Of course, our university has an Ethics Committee. But it refused to satisfy us, because these are not direct studies on humans or animals.
- What is chemical composition of aligners?
We used Nuvola® aligners. They are made of polyethylene terephthalate glycol (PETG), a light, resistant, and very clear material.
Do its composition have influence on type of biofilm growth?
May be) Next time we will do the research with Fantasmino® aligners, which are made of poly-vinyl chloride (PVC)
- It would be interesting if authors bifurcate data in table on gender basis.
We will try to take this into account in the following experiments. Thanks for the idea. I'm afraid it's not possible right now
- Please comments on dominance of YLF in sampled biofilms. Do sampling time, type of aligners material, diet or age group have any role to play in it?
Thank you for paying attention to this. Most of our previous works are devoted to candida. We have given a detailed explanation of this phenomenon on page 4:
- albicans' colonization of the mucous membranes may result in the acquisition and preservation of a stable population of Candida, preventing clinical infection de-velopment. The higher YLF concentration can be explained by….
- Confocal laser scanning microscopy images and interpretation is missing.
Of course, because we had a negative result. Should we add photos in black only? There was no glow) Malevich «Black Square»))))
- Conclusion need revision. Explain how your data is helpful in new drug development?
In the future, we plan to use the obtained strains of microorganisms to create an effective and safe biofilm-destroying agent.
Thank you very much, we will redo the structure of the article with your edits
Reviewer 3 Report
Introduction should focus more on the strains involved as well as on biofilm formation
Materials and methods
2.1. Microbial strains
This chapter has to be re-written almost entirely.
State the protocol in a clear, step by step manner.
"If different scores were detected by MALDI Biotyper" If MALDI-TOF analysis was performed, please state protocol and data base used
Separate Results from Discussions for clarity of the results
I suggest that the authors first describe the methodology properly and then clearly state the results before heading to the discussions.
Author Response
Dear Reviewer! Thank you so much for paying attention to our work and spending your time. Our team very much appreciates your edits in the article and of course we will take them into account. We are sure that working together will only make the article better. We have tried to answer all your questions:
2.1. Microbial strains
This chapter has to be re-written almost entirely.
State the protocol in a clear, step by step manner.
In section 2.1, we wrote that we took the used aligners from the patient: from the lower and upper jaw. Microbiological analysis was performed according to the international protocol. The incubation of microorganisms lasted from 24 to 72 hours. The biofilms were incubated for 48 hours. In the methodology, we described the incubation conditions and the mediums. We also provide links to articles previously published by us, where everything is described in detail. We can't duplicate our work the same way every time.
"If different scores were detected by MALDI Biotyper" If MALDI-TOF analysis was performed, please state protocol and data base used
We used established guidelines CLSI for microbiological identification by MALDI-TOF MS (https://clsi.org/standards/products/microbiology/documents/m58/, accessed on 6 Desember 2022).
Both databases of the two main MALDI-TOF MS systems currently available, MALDI Biotyper (Bruker Daltonics GmbH, Bremen, Germany) and VITEK MS/MS Plus (bioMérieux, Marcy l’Etoile, France), include common yeasts isolated in clinical microbiological laboratories. But we used the first one.
Separate Results from Discussions for clarity of the results
I suggest that the authors first describe the methodology properly and then clearly state the results before heading to the discussions.
Thank you very much, we will redo the structure of the article with your edits
Round 2
Reviewer 1 Report
Dear Authors,
thank you very much for your work.
I think that all my comments have been considered for the improvement of the manuscript.
I thus think that the manuscript is now suitable for publication in the journal.
Yours faithfully.
Author Response
Dear reviewer. Thank you so much for your help in editing the article. Your comments are very valuable to us. Our team wants to wish you good luck and health in the New Year! Happy New Year and Merry Christmas!!!!
Reviewer 2 Report
Authors have adressed most of the comments. But still their is room for improvement in text and clarification of imbiguties. Authors should carefully revise the final manuscript to meet the required standards of publication.
Author Response

(The authors gave the same response as above.)

Reviewer 3 Report
Baybekov et al. improved their manuscript considerably but some aspects were ignored.
"Introduction should focus more on the strains involved as well as on biofilm formation"
No modifications were made to this chapter nor an explanation why. Some data on this topic is presented in discussions. Please provide relevant data regarding biofilms under introduction and expand them under discussions based on the results obtained.
Results and discussions were kept together, if separating them is not preferred I have nothing against such structure.
Author Response
Dear reviewer. Thank you so much for your help in editing the article. Your comments are very valuable to us. Our team wants to wish you good luck and health in the New Year! Happy New Year and Merry Christmas!!!!
Let us answer your questions:
"Introduction should focus more on the strains involved as well as on biofilm formation". No modifications were made to this chapter nor an explanation why. Some data on this topic is presented in discussions. Please provide relevant data regarding biofilms under introduction and expand them under discussions based on the results obtained.
Since this study is the first of a whole series of studies, we decided not to delve into this subject in the first article. In the next article there will be a different age of the participants and there in the discussion we will focus on strains and differences in biofilms.
Results and discussions were kept together, if separating them is not preferred I have nothing against such structure.
Let us leave the Discussion and the Result united together. Thank you!